# Handgrip Strength Is Positively Associated with 24-hour Urine Creatine Concentration

**DOI:** 10.3390/ijerph20065191

**Published:** 2023-03-15

**Authors:** Enkhtuya Ulambayar, Delgermaa Bor, Nandin-Erdene Sukhbaatar, Narkhajid Usukhbayar, Uugantuya Ganbold, Odmaa Byambasuren, Uranbaigali Enkhbayar, Oyuntugs Byambasukh

**Affiliations:** 1Department of Clinical Laboratory, School of Medicine, Mongolian National University of Medical Sciences, Ulaanbaatar 14210, Mongolia; 2Department of Endocrinology, School of Medicine, Mongolian National University of Medical Sciences, Ulaanbaatar 14210, Mongolia; 3Department of Nephrology and Endocrinology, Central Military Hospital, Ulaanbaatar 13341, Mongolia; 4MONPRD Research Team, MASO, Ulaanbaatar 14210, Mongolia

**Keywords:** muscle strength, creatinine excretion, grip strength, sarcopenia

## Abstract

Background: Muscle mass evaluation methods are often expensive and therefore limited in their daily use in clinical practice. In this study, we investigated the relationship between hand grip strength (HGS) and other parameters of body measurements with urine creatinine, especially to investigate whether HGS measurement is an indicator of muscle metabolism. Methods: In total, 310 relatively healthy people (mean age 47.8 + 9.6; 161 people or 51.9% of the total population were men) who were undergoing preventive examinations were included in this study and given a container to collect 24-h urine, and the amount of creatinine in the urine was determined by a kinetic test without deproteinization according to the Jaffe method. A digital dynamometer (Takei Hand Grip Dynamometer, Japan) was used in the measurement of HGS. Results: There was a significant difference in 24-h urine creatinine (24 hCER) between the sexes, with a mean of 1382.9 mg/24 h in men and 960.3 mg/24 h in women. According to the correlation analysis, the amount of urine creatinine was related to age (r = −0.307, *p* < 0.001 in men, r = −0.309, *p* < 0.001 in women), and HGS (r = 0.207, *p* = 0.011 in men, r = 0.273, *p* = 0.002 in women) was significant for either sex. However, other parameters of body measurements, such as girth, forearm circumference, and muscle mass measured by bioelectrical impedance, were not related to urine 24 hCER. A correlation between HGS and 24 hCER was observed in age groups. Conclusions: We found that HGS is a potential marker in muscle metabolism assessment that is proven through 24 hCER. In addition, therefore, we suggest using the HGS measure in clinical practice to evaluate muscle function and well-being.

## 1. Introduction

Handgrip strength (HGS) is an important health indicator for older adults, as its lower value is associated with adverse outcomes such as fragility and falls [1,2]. As described in a review study, a lower HGS is positively associated with the risk of fragility independently with various aging markers and physical functioning [1]. A recent prospective study showed that a greater HGS was associated with better functional outcomes after 1-year follow-up among fragility hip fracture patients [3]. In addition, falls and fracture risk are greater among seniors with lower HGS [3,4]. A meta-analysis (n = 220,757) of 19 population-based prospective cohort studies showed a significant correlation between hand grip strength and the predictability of bone fractures, and the odds ratio was 0.70 in the highest tertile group compared to the lowest tertile group of HGS, which shows that the fracture risk was low in people with higher HGS [4]. Furthermore, recent studies show that the risk of mortality is higher in individuals with low HGS. For instance, in a Spanish study of 351 hospitalized cancer patients, the risk of mortality was not associated with body circumference measures, while the HGS measurement proved to be a significant predictor of mortality [5]. In recent years, in particular, clinical guidelines and recommendations have included HGS as a muscle strength indicator in the criteria for identifying lower muscle mass or sarcopenia [6,7,8,9].

In the literature, creatinine is an intermediate product of muscle metabolism and is regularly excreted in the urine at a rate of 1 g/day under normal conditions [10]. Most of the creatinine is produced by non-enzymatic metabolism, which occurs at a steady rate in the skeletal muscle as a function of muscle mass [11,12]. Therefore, urinary creatinine excretion is the main determining factor of skeletal muscle metabolism. Although measuring serum creatine is important in evaluating muscle metabolism, researchers noted that measuring creatinine in urine, particularly in urine samples taken 24 h a day (24 hCER), is an appropriate method for assessing muscle metabolism [13,14,15]. A few studies have examined the relationship between HGS and urinary creatinine excretion; most have been conducted in patients with chronic illness [16,17,18]. 

It is important to investigate an easily accessible and inexpensive tool to measure muscle metabolism; one would be HGS. In this study, we hypothesize that HGS could be a potential marker for muscular metabolism that can be shown to possess a positive association with urinary creatine excretion. Therefore, we aimed to explore the relationship between HGS and 24 hCER among healthy adults. We also investigated whether HGS is a better indicator of muscle metabolism than other body composition measures in relation to 24 hCER.

## 2. Materials and Methods

### 2.1. Study Participants

This study was conducted among those who participated in health screening at the university hospital of the Mongolian National University of Medical Sciences between July and September 2022, and those who agreed to participate in the study (n = 443). The sample size was calculated based on the total number of people screened for the last 6 months in the hospital (n = 12,500) and assuming a 95% confidence interval (Z = 1.96) with a 5% acceptable margin of error (e = 0.05), which gave a sample size of 443 persons. During the data collection, we excluded individuals who were being treated for chronic diseases, which can impact muscle metabolism, such as diabetes and liver cirrhosis, as stated earlier in the patient report (n = 90). We also excluded cases where these conditions were newly diagnosed by doctors as a result of health screening (n = 19). Moreover, if an athlete had participated in sports sessions in the last seven days or if patients were using diuretics, we excluded them (n = 7). In addition, individuals who did not collect urine more than twice in the 24-h urine collection period and those whose measurement was less than 500 mL/24 h were excluded (n = 19) from the study [19]. A total of 310 participants were included in the current analysis. 

The study was conducted according to the Helsinki Declaration, and it was approved by the medical ethical committee of the Mongolian National University of Medical Sciences (METc 2022/Z-03). All participants provided their written informed consent.

### 2.2. Hand Grip Strength Measurement

A digital dynamometer (Takei Hand Grip Dynamometer 5401-C, Tokyo, Japan), which is an isometric electronic device, was used to measure HGS in this study [20]. The instrument weighed 5 to 100 kg, and the minimum unit of measurement was 0.1 kg. The HGS was measured two times for both hands and the mean for each hand was recorded. During the measurement, the participant was recorded with feet shoulder-width apart, elbows fully extended, and fingers bent 90° once in each hand. Participants were instructed to continually tighten the handle with full force for at least 3 s and not to move the dynamometer or hold their breath while taking the measurement. The measures were the same for men and women and age groups, but we used different thresholds that had been established before in the same population [21]. The maximal measurement for one of the two hands was taken and called the dominant HGS [20].

### 2.3. 24-h Urine Creatinine Collection and Measurement

The kinetic test without deproteinization according to the Jaffe method was used to measure urine creatinine (BioMajesty^®^ BM6010/C) [22]. Subjects were provided with a designated urine collection container and advised to stay at home or choose a convenient day for urine collection. When providing instructions, it was recommended that urine collection began at 8:00 a.m. or the time and date at the start of urine collection was recorded. In addition, on the day of urine collection, it was recommended not to consume beets, coffee, or foods that were too sweetened; not to use diuretics; and not to perform physical workouts and heavy sports.

### 2.4. Other Variables and Measurements

The interview included questions related to participants’ general and lifestyle characteristics, including education, smoking, alcohol use, diet, and physical activity. Questions on the intake of meat, fruits, and vegetables were based on a daily to weekly frequency. We assessed physical activity behavior, using a question that dealt with workouts, exercise, and sports [23]. The question was “How often do you perform physical activities to work out or exercise or sports?” The answers were “never or less than once monthly”, “1–2 times per month”, “1–2 times per week”, “3–4 times per week”, and “daily”. Based on the descriptive results, lifestyle variables were classified as dichotomous variables: smokers/non-smokers, weekly/non-weekly alcohol use, daily/non-daily intake of meat, fruit and vegetable intake, and weekly/non-weekly regular physical activity. 

Participants’ body weight (in kg), height (in cm), blood pressure (Pangao, PG-800B69, The Hague, Netherlands), and body circumferences, such as neck, chest, midarms (upper arm circumference), forearm (lower arm circumference), waist, hip, and thigh circumference, were measured by well-trained assistants implementing a standardized protocol. Body mass index (BMI; kg/m^2^) was subsequently calculated based on body weight and height. A bioelectric impedance analyzer (Tanita BC-730) was used for muscle mass measurement (in kg). 

### 2.5. Statistical Analysis

The characteristics of the study population were expressed as means with standard deviation (SD) and as numbers with percentages according to 24-h urine creatinine tertiles. A histogram of the 24-h urine creatinine concentration was obtained and expressed as a median with a 25th–75th percentile. The differences between groups were compared using Student’s *t*-test, one-way analysis of variance (ANOVA), the Kruskal–Wallis test, and Pearson’s Chi-Square test. 

Crude and age-adjusted Spearman’s correlation coefficient was calculated between 24-h urine creatinine and age, HGS, and other anthropometry measures. Furthermore, sex-stratified and age-adjusted estimation of the 24-h urine creatinine concentration was performed in stratified groups of age, physical activity, and HGS category using univariate analysis. The age group was determined based on age tertiles. The HGS category was based on previous study results that categorized dominant HGS into higher and less than the 25th percentile of HGS based on its histogram in men and women, respectively. Less than the 25th percentile of HGS (31.8 for men and 20.5 for women) was categorized as the lower HGS group [21]. Additionally, a linear regression analysis was performed, and the unstandardized beta coefficient was reported with a 95% confidence interval (CI). After crude analysis, the analysis was adjusted for age, education, and BMI.

For all statistical analyses, we used IBM SPSS V.28.0. A statistical significance level was set at *p* < 0.05 for all tests.

## 3. Results

The mean age of the subjects was 47.8 + 9.6 years and 51.9% (n = 161) of the total subjects were male. The amount of creatinine excreted in the urine was significantly different for the two genders, with a mean of 1382.9 mg/24 h (median of 1322.0 and 25th–75th percentile of 869–1815) in men and 960.3 mg/24 h (median of 878 and 25th–75th percentile of 583–1265) in women. Therefore, further analyses were performed by gender.

Table 1 shows the characteristics of the study participants, based on urine creatinine levels (tertiles). Significant differences were observed for age and HGS, and it seems that with increasing age, urinary creatinine levels declined, while those with high HGS tended to have higher creatinine levels in the urine. It was assumed that meat consumption, physical activity frequency, and muscle mass measured by the bioelectrical impedance analyzer were consistent with creatinine levels in the urine, but no differences were observed between the 24 hCER groups. Regarding the level of education, 13% of men and 10% of women had low education and no differences between groups were found (*p* > 0.05). In addition, there were no differences between groups (24 hCER groups) in systolic blood pressure.

As shown in Table 1, there was a tendency for urine creatinine to increase with increased physical activity, but this was not significant. Therefore, age-adjusted urine creatinine was compared for three levels of physical activity. Although there was a tendency for urine creatinine to increase with an increased level of physical activity, this was not significant. 

Spearman’s correlation analysis showed that the 24 hCER was correlated significantly only with age and HGS in both sexes (Table 2). Although no significant correlation was observed, we can see from the table that there was a correlation between 24 hCER and BMI and muscle mass determined by the bioelectric impedance analyzer. For other body composition measures, body height and weight alone, as well as body circumference measurements, were not related to urine creatinine excretion. In addition, when adjusting for age, the correlations of 24 hCER with HGS remained.

The positive correlation between 24 hCER and HGS shown in the table above shows that people with higher HGS or muscle mass may possess a higher metabolism of creatinine as they present higher 24 hCER (Figure 1).

In the additional analysis, when we used the reference value of the study that established the lower limit of HGS among Mongolian people (31.8 for men and 20.5 for women), Figure 2 shows that with increasing age, the number of people in the low HGS group increases. 

When we used the HGS groups mentioned above, there were significant differences in the amount of creatinine excreted in the urine by groups of HGS (Table 3). Moreover, when considering age groups, the amount of creatinine excreted in the urine tended to decrease. Considering both sexes, 24 hCER levels were significantly lower in men with increasing age in the lower HGS groups. Similarly, in women, 24 hCER decreased in the low HGS groups with increasing age, but only a significant difference was observed in the middle age group (Table 3).

Finally, we tested the relationship between HGS and 24 hCER using linear regression analysis (Table 4). The HGS was significantly associated with 24 hCER in both men and women. After adjusting for age, the association was attenuated but remained significant, indicating that the association was age-independent. Further adjustments showed the significant association of HGS with 24-h urine creatinine. 

## 4. Discussion

We found that there was a significant association between HGS and the level of creatinine in the urine within 24 h. We hypothesized that other body composition measures, especially muscle mass measured by bioelectrical impedance analysis, were positively associated with 24 hCER, but there were no significant correlations observed. These suggest that HGS may be a better indicator of muscle mass compared to other body composition indices, because urine creatinine is the main determining factor in skeletal muscle metabolism. Furthermore, there were significant differences in 24 hCER by sex and age, but the association of HGS with 24 hCER was independent of age and sex.

Most previous studies examining the relationship between HGS and urinary creatinine levels were linked to nutritional status or in individuals with chronic renal disease [16,17,18]. A study in the Netherlands (including 184 renal transplant recipients) showed a significant association between HGS and 24 hCER (standardized β = 0.33, *p* < 0.001), independent of adjustment for potential confounders such as age, sex, eGFR, time after transplantation, living donor, BSA, history of CVD, hypertension, glucose levels, albumin, lipids, hs-CRP medication use, and protein intake [17]. In 50 dialysis patients, the rate of endogenous creatinine synthesis was calculated and found to be positively associated with HGS (β = 0.44, *p* < 0.001) [18]. These studies also tested the association of HGS with other muscle characteristics, including walking tests, etc., showing that HGS is a useful measure for assessing muscle function, not merely muscle strength. We found a significant association between HGS and urinary creatinine levels within 24 h in relatively healthy individuals. These suggest that this simple and fast method is suitable for clinical use, as HGS can be used as an indicator of muscular metabolism, in relatively healthy people and people with chronic illnesses.

Although we found that there were no associations of other body composition measures with urine creatinine, previous research has found anthropometric predictors of creatine clearance in urine, including body weight, height, and girth [24,25,26,27,28]. Joachim et al. examined the association of the urinary creatinine excretion rate with mortality risk in a relationship with coronary artery disease [26]. The study analyzed the associations of body circumferences and the calculated rate of creatinine excretion (CER) with a marker of muscle mass. Only the WHR had a weak association with CER. Barrios et al. analyzed the relationships between waist circumference (WC), waist–hip ratio, and 24-h urinary albumin excretion rate (UAER) and creatine clearance [27]. Study participants underwent measurements of anthropometric factors and calculations (including WC, waist to hip ratio) and 24-h urine sampling to determine creatinine clearance and UAER. Of the anthropometric parameters, one increase in WC was found to be independently linked to 24 h UAER, although this association disappeared after the correction of creatinine clearance on the body surface area [28]. In addition, the researchers developed some indices, such as the creatine height index and the creatine arm index [29,30]. However, these indices are not commonly used for clinical application or research. Moreover, not all studies found a significant association between urinary creatinine and the aforementioned body composition indices [16,18]. It could be explained that these indexes are not direct measures of muscle mass or function. Indeed, HGS is also not a direct measure of muscle metabolism. Our results suggest that HGS might be a better measure than other anthropometric measures. Furthermore, some studies have shown a positive association between urinary creatinine and muscle mass measured by bioelectrical impedance analyzers (BIA) [16]. Our study did not reveal any significant association among these variables. This may be explained by the capacity and estimation of the BIA methodology. The results of a bioelectric impedance analysis are dependent on the BIA equation and the number of sensors used in the measurement [31]. For instance, one study found a significant association between 24 hCER and HGS, using the multifrequency BIA method [16]. A single-frequency BIA analyzer was used for this study. Therefore, we suggest using HGS in muscle evaluations instead of the single-frequency BIA analyzer in clinical settings.

In accordance with the literature, we found that the concentration of urinary creatinine depends on age and gender [10,24,32]. As a result, the current study has identified creatinine differences by HGS with an age effect. The participants were divided into two groups, those with high HGS (>25 percentile) and low HGS (<25 per percentile); this threshold criterion was found in a previous study in the same population [21]. There were certain associations for the urinary creatine concentration in relation to HGS groups and age groups in this study. The relations among age–HGS and age–concentration of urinary creatine were logically correlated in males. The incidence of lower HGS and age were positively correlated, and a low urinary level of creatinine could be associated with sarcopenia. However, not all associations in the age groups were relevant for women. This may be associated with lower levels of creatine in urine in women than in men in order to divide them into subgroups.

Studies showed that the urinary level of creatinine was induced by physical activity and exercise [16,33]. In the present study, there was no relationship between them. This could be due to the exclusion criterion used in the study that excluded people who had exercised in the last three days. In addition, studies have shown that creatinine in the urine is linked to protein intake [34,35]. There was no correlation between meat consumption and 24 hCER in this study. This could be explained by the fact that Mongolians regularly consume a large amount of meat, which does not lead to any difference [36].

According to the research hypothesis in this study, HGS seems to be a good marker for muscle metabolism, as shown by a positive association with urinary creatinine excretion. Therefore, HGS measurement should be used in daily clinical practice to assess muscle function and well-being, particularly in older adults, to prevent frailty and falls.

There was a limitation in our study. The sample was not sufficient to conduct age group analyses for other groups of variables, such as sex, 24 hCER, and HGS. Furthermore, we did not calculate eGFR and creatinine clearance using the serum creatinine level in our study. Finally, the participants in this study were relatively young.

## 5. Conclusions

We found that HGS is a potential marker in muscle metabolism assessment that is proven through 24 hCER. In addition, HGS can be a better indicator of muscle mass compared to other body composition indexes, such as single-frequency BIA and body circumferences. Therefore, we suggest using the HGS measure in clinical practice to evaluate muscle function and well-being.

## Figures and Tables

**Figure 1 ijerph-20-05191-f001:**
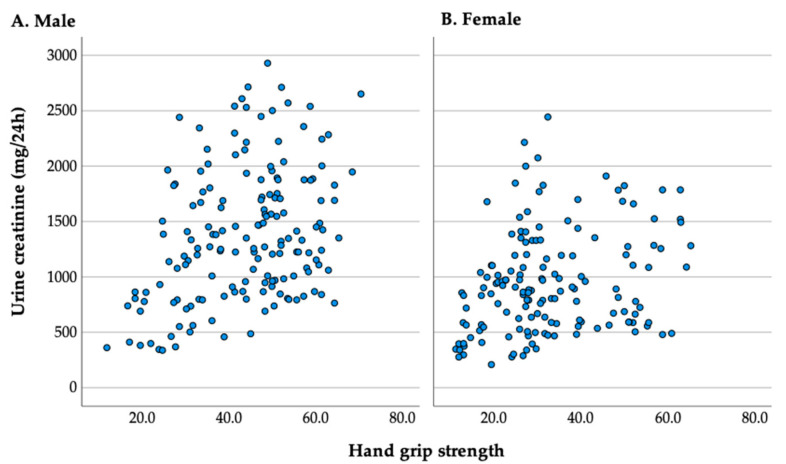
HGS and 24 hCER, according to sex.

**Figure 2 ijerph-20-05191-f002:**
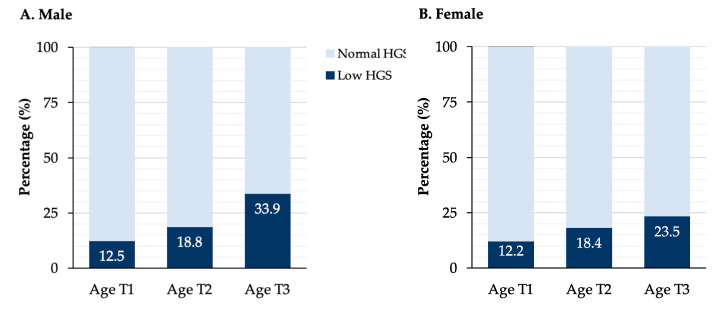
Percentage of low HGS according to age groups in men and women.

**Table 1 ijerph-20-05191-t001:** Characteristics of study population.

Findings	Total	24-h Urine Creatinine Level			*p*-Value
		T1	T2	T3	
**Male (n)**	**161**	**53**	**54**	**54**	
Age (years)	42.5 ± 9.6	46.8 ± 10.8	41.1 ± 8.6	39.8 ± 7.8	**0.002**
Height (cm)	169.0 ± 8.2	168.1 ± 8.3	169.4 ± 7.5	169.6 ± 8.7	0.551
Weight (kg)	78.7 ± 14.1	77.9 ± 14.1	77.3 ± 14.2	80.9 ± 13.9	0.364
BMI (kg/m^2^)	28.9 ± 17.0	27.3 ± 3.4	27.0 ± 3.7	32.3 ± 28.7	0.144
Systolic blood pressure (mmHg)	114.5 ± 12.9	116.1 ± 11.7	115.4 ± 12.1	111.9 ± 13.4	0.401
Muscle mass (kg)	53.5 ± 38.3	50.0 ± 10.5	57.9 ± 65.1	52.5 ± 10.3	0.372
Neck circumference (cm)	39.2 ± 5.0	38.1 ± 4.0	39.8 ± 7.1	39.6 ± 3.0	0.054
Waist circumference (cm)	92.3 ± 10.9	92.0 ± 10.4	90.5 ± 11.7	94.5 ± 10.3	0.240
Hip circumference (cm)	100.3 ± 8.3	101.0 ± 8.1	100.0 ± 8.5	99.9 ± 8.5	0.741
Upper arm circumference (cm)	33.5 ± 25.9	31.2 ± 4.1	37.6 ± 4.2	31.6 ± 3.9	0.589
Lower arm circumference (cm)	23.5 ± 4.6	23.4 ± 4.5	23.5 ± 4.6	23.7 ± 4.7	0.839
Tigh circumference (cm)	54.7 ± 6.4	53.6 ± 6.1	54.6 ± 6.7	56.0 ± 6.3	0.108
Hand grip strength (kg)	43.7 ± 12.8	38.0 ± 14.1	45.6 ± 11.3	47.5 ± 11.1	**0.002**
Meat intake (daily)	98.1 (158)	98.1 (52)	96.3 (52)	100.0 (54)	0.363
Fruit and vegetable intake (daily)	5.0 (8)	5.7 (3)	1.9 (1)	7.4 (4)	0.398
Regular physical activity (weekly)	42.2 (68)	41.5 (22)	38.9 (21)	46.3 (25)	0.732
Education (lower level)	13.0 (21)	11.3 (6)	14.8 (8)	13.0 (7)	0.801
Alcohol use (weekly)	24.8 (40)	35.8 (19)	18.5 (10)	20.4 (11)	0.075
Smokers	55.3 (89)	47.2 (25)	53.7 (29)	64.8 (35)	0.178
**Female (n)**	149	49	50	50	
Age (years)	45.1 ± 9.4	49.8 ± 9.8	44.1 ± 8.6	41.6 ± 7.9	**<0.001**
Height (cm)	164.6 ± 8.2	165.0 ± 8.3	164.0 ± 8.5	164.8 ± 7.9	0.790
Weight (kg)	73.2 ± 12.9	75.4 ± 13.1	72.0 ± 12.2	72.4 ± 13.5	0.366
BMI (kg/m^2^)	26.9 ± 3.7	27.6 ± 3.6	26.8 ± 3.7	26.4 ± 3.7	0.227
Systolic blood pressure (mmHg)	117.5 ± 11.4	118.3 ± 9.9	116.1 ± 11.9	118.1 ± 10.9	0.401
Muscle mass (kg)	45.7 ± 9.5	46.8 ± 10.1	44.8 ± 8.5	45.7 ± 9.9	0.867
Neck circumference (cm)	36.7 ± 3.7	37.3 ± 3.2	36.4 ± 4.4	36.3 ± 3.3	0.331
Waist circumference (cm)	88.1 ± 12.0	91.2 ± 11.6	86.3 ± 11.5	87.0 ± 12.5	0.094
Hip circumference (cm)	99.2 ± 8.2	100.7 ± 7.6	98.1 ± 8.1	98.6 ± 8.8	0.231
Upper arm circumference (cm)	30.2 ± 3.6	30.7 ± 3.5	29.9 ± 4.0	30.1 ± 3.3	0.245
Lower arm circumference (cm)	23.6 ± 4.0	24.0 ± 3.7	23.7 ± 3.7	22.9 ± 4.6	0.322
Tigh circumference (cm)	53.1 ± 6.1	52.4 ± 6.2	53.6 ± 6.3	53.4 ± 5.7	0.503
Hand grip strength (kg)	33.0 ± 13.8	30.8 ± 14.8	29.9 ± 10.9	38.2 ± 13.9	**0.009**
Meat intake (daily)	98.0 (146)	95.9 (47)	98.0 (49)	100.0 (50)	0.352
Fruit and vegetable intake (daily)	4.0 (6)	4.1 (2)	6.0 (3)	2.0 (1)	0.596
Regular physical activity (weekly)	37.2 (55)	30.6 (15)	40.0 (20)	40.8 (20)	0.508
Education (lower level)	10.1 (15)	8.2 (4)	12.0 (6)	10.0 (5)	0.782
Alcohol use (weekly)	18.1 (27)	22.4 (11)	14.0 (7)	18.0 (9)	0.694
Smokers	38.9 (58)	34.7 (17)	40.0 (20)	42.0 (21)	0.744

Data are presented as mean ± SD and percentages, % (number). Bold values denote statistical significance at the *p* < 0.05 level.

**Table 2 ijerph-20-05191-t002:** Association of 24-h urine creatinine with body composition indices.

24 hCER	Male		Female	
	r	*p*-Value	r	*p*-Value
Age	−0.307	**<0.001**	−0.309	**<0.001**
HGS	0.207	**0.011**	0.273	**0.002**
BMI	0.097	0.237	−0.037	0.680
Muscle mass	0.018	0.831	−0.016	0.857

Data are presented as Spearman’s correlation coefficients. Bold values denote statistical significance at the *p* < 0.05 level.

**Table 3 ijerph-20-05191-t003:** Mean 24 hCER according to sex, age, and HGS category.

Age Groups	Total	HGS Category		
		Normal	Low	*p*-Value
**Male**				
Age tertile 1	1556.0 ± 540.1	1574.0 ± 541.0	1434.1 ± 554.2	0.528
Age tertile 2	1507.0 ± 649.0	1620.1 ± 589.2	1020.1 ± 705.3	**0.011**
Age tertile 3	1115.1 ± 517.2	1297.1 ± 516.2	759.1 ± 288.3	**<0.001**
Total	1387.1 ± 598.9	1506.2 ± 562.3	961.2 ± 534.1	**<0.001**
**Female**				
Age tertile 1	1138.1 ± 566.4	1184.3 ± 562.9	805.3 ± 521.9	0.126
Age tertile 2	1014.5 ± 436.1	1095.4 ± 424.4	653.2 ± 291.4	**0.005**
Age tertile 3	739.4 ± 367.8	776.1 ± 385.1	619.4 ± 278.3	0.199
Total	960.1 ± 489.3	1024.2 ± 495.5	672.2 ± 341.7	**<0.001**

Data are presented as mean ± SD. Bold values denote statistical significance at the *p* < 0.05 level.

**Table 4 ijerph-20-05191-t004:** Association of HGS and 24-h urine creatinine.

HGS	Unstandardized Beta Coefficient of 24 hCER			
	Male		Female	
	β (95% CI)	*p*-Value	β (95% CI)	*p*-Value
Crude	16.1 (9.3–22.9)	**<0.001**	8.6 (3.3–13.9)	**0.002**
Adjusted for age	12.5 (5.3–19.6)	**<0.001**	8.1 (3.1–13.1)	**0.002**
Adjusted for age and education	11.1 (3.9–18.1)	**0.002**	8.2 (3.2–13.3)	**0.002**
Adjusted for age and BMI	12.1 (5.1–19.0)	**<0.001**	8.2 (2.9–13.1)	**0.002**

Data are presented as unstandardized beta coefficient with 95% CI. CI, confidence interval. Bold values denote statistical significance at the *p* < 0.05 level.

## Data Availability

The data used to support the findings of this study are available from the corresponding author upon request.

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
