# Peer review of "Handgrip Strength Is Positively Associated with 24-hour Urine Creatine Concentration"

_ijerph, 2023, doi:10.3390/ijerph20065191_

Round 1

Reviewer 1 Report

Thank you for the opportunity to review, I have the following comments:

·        L40-41 - The sentence, to me, does not show a cause and effect relationship. Please elaborate on this thought.

·        L43 – ‘’ markers and physical functioning [1-2]’’ - The authors refer to quite old studies from 2010 and 2003. I suggest updating the literature and references.

·        At the end of the discussion, I suggest adding a research hypothesis.

·        L71 – ‘’ MNUMS’’ - Please elaborate on the name.

·        L72 -73 -Who diagnosed and conducted the study in terms of inclusion and exclusion criteria?

·        L78 – ‘’ 310’’ -  Please add information on how to count the sample size.

·        L80-82 and L283-285 - Please provide the exact number of bioethics committee approval.

·        L83 - Do I understand correctly that only one compression force measurement was made?

·        The conclusions are too much repetition of the results. Please reword.

·        Note to all the text, when quoting, you only need to give the name without the initial of the first name. For example (L225) change ‘’ Barrios A’’ to ‘’ Barrios’’. Please correct throughout the text.

·        References in their entirety should be corrected according to the requirements of the journal.

Author Response

Thank you very much for giving us the opportunity to improve and resubmit our manuscript. We appreciate your helpful comments and suggestions to strengthen the quality of our manuscript. We are submitting this revised manuscript after carefully considering your useful comments. 

Reviewer 2 Report

Thank you for having an opportunity to read this manuscript. The research purpose and the results were interesting and clearly presented. However, there were some missing information and concerns as indicated below. Please add to explain the reason clearly why HGS is important to be investigated.

In the Results, the “education” was mentioned (Table 4). Please add the explanation in the Methods (2.4).

Abstract

Line 21, Line 26, Line 33: (1), (2), & (3) should be erased. Any reason to be put?

Introduction

Line 67: Hypothesis should be added.

Line 70-78: Participant numbers were different from the following number calculation. 443-90-18-7-19=309. However, the participants were 310. 

Line 82: Approval number should be added.

Methods

Line 83: How many times was the HGS measured?

Line 114: The results of blood pressure were not provided in the Results.

Line 126: ANOVA is abbreviated. Please correct without abbreviation.

Table 1: Please add data of blood pressure too. 

The unit of the Meat intake (grams?)

Line 266: Why did the authors consider an efficient sample was used?

Author Response

(The authors gave the same response as above.)

Reviewer 3 Report

This article is very interesting. It deals with the important topic of handgrip strength is positively associated with 24-hour urine creatine concentration.

To the authors.

Every part of this article is very clear, but please modify some contents.

1. Please delete serial number in abstract.

2. P values need to be italicized 

3. Please add the contents that why the relevance between handgrip strength and urine creatinine concentration is a valuable issue in introduction. 

4. Please add the content of implication or perspective in discussion to highlight the application value of this research.

Author Response

(The authors gave the same response as above.)
